

# The critical need for child and youth perceptions of active living in India: capturing context complexity in rural and urban regions

Tarun Reddy Katapally[1,2,3,4], Jamin Patel[1,2], Anuradha Khadilkar[4] and Jasmin Bhawra[4,5]

[1] DEPtH Lab, Faculty of Health Sciences, University of Western Ontario, London, Ontario, Canada
[2] Department of Epidemiology and Biostatistics, Schulich School of Medicine and Dentistry, University of Western Ontario, London, Ontario, Canada
[3] Children's Health Research Institute, Lawson Health Research Institute, London, Ontario, Canada
[4] Hirabai Cowasji Jehangir Medical Research Institute, Pune, Maharashtra, India
[5] CHANGE Research Lab, School of Occupational and Public Health, Toronto Metropolitan University, Toronto, Ontario, Canada

Corresponding author
Tarun Reddy Katapally,
tarun.katapally@uwo.ca

## ABSTRACT

**Background**. The physical inactivity pandemic not only has a negative impact on the physical and mental health of children and youth, but it is also a key contributor to the non-communicable disease (NCD) burden, particularly among low- and middle-income countries. The widespread effects of climate change, ranging from extreme weather events to worsening air quality, are exacerbating the physical inactivity pandemic, highlighting the need to undertake holistic interventions to address environmental barriers while promoting physical activity. Despite the potential benefits of active school transportation (AST) on physical activity and the environment, no study has examined the intersection between perceptions of AST, environmental perceptions, and moderate-to-vigorous physical activity (MVPA) among children and youth in India from a child and youth perspective.

**Methods**. As a part of this cross-sectional, observational study, digital survey links were administered to 1,042 children and youth aged 5 to 17 (50.3% boys; 49.7% girls). Participants were recruited from 41 schools across 28 rural and urban regions in India. Children and youth provided information on various sociodemographic factors, perceptions of crime and air pollution, and peer support factors. MVPA data were collected using an adapted version of the IPAQ short-form questionnaire. The overall sample was stratified by age, gender, and location, resulting in a total of seven multiple linear regression models to assess the association between AST and MVPA.

**Results**. Children and youth who engaged in AST were associated with higher MVPA than those who did not engage in AST ($\beta = 18.020$, 95% CI [5.890 to 30.149], $p = 0.004$), after adjusting for age, gender, and location. In contrast, perceptions of high crime in the neighbourhood ($\beta = -21.921$, CI [−36.195 to −7.647], $p = 0.003$) and perceptions of air pollution ($\beta = -12.472$, CI [−23.797 to −1.147], $p = 0.031$) were associated with lower MVPA. Moreover, having active friends was associated with higher MVPA ($\beta = 32.391$, CI [9.264 to 55.518], $p = 0.006$) than not having active friends. AST was significantly associated with higher MVPA in the boys, rural, and

aged 13 to 17 models; however, this association was not statistically significant in the girls, urban, and aged 5 to 12 models.

**Conclusions**. These findings highlight that promoting AST can potentially promote MVPA and contribute to mitigating the NCD burden among children and youth in India. Future policies and interventions should prioritize initiatives that promote AST, considering diverse sociodemographic factors, and addressing environmental challenges such as perceptions of crime and air pollution.

## INTRODUCTION

Non-communicable diseases are one of the leading causes of death worldwide and have become a significant global public health challenge, particularly in low-and middle-income countries where they account for seven out of every ten deaths (*Ndubuisi, 2021*). In India, the burden of non-communicable diseases has exponentially increased due to urbanization and population growth. For instance, a study conducted in Tamil Nadu, a southern Indian state, found urbanization to be linked to the prevalence of several non-communicable disease risk factors, including lower physical activity (*Allender et al., 2010*).

Given that child and youth health behaviours often track into adulthood (*Due et al., 2011*), it is imperative to take proactive measures that address the numerous interconnected factors contributing to non-communicable diseases (*Hadian et al., 2021*). Promoting physical activity can be an important proactive measure to mitigate the global non-communicable diseases burden as regular physical activity contributes to overall health and well-being, particularly among children and youth (*Poitras et al., 2016*; *Granger et al., 2017*; *Marker, Steele & Noser, 2018*). In particular, the World Health Organization recommends at least an average of 60 min of moderate-to-vigorous physical activity (MVPA) per day for children and youth, with other research making similar recommendations based on the available evidence (*WHO, 2020*; *Janssen & LeBlanc, 2010*). Meeting the MVPA guidelines has not only been associated with reduced risk of non-communicable diseases, but it also provides several other health benefits, including improved cardiovascular fitness and musculoskeletal strength (*Poitras et al., 2016*; *Janssen & LeBlanc, 2010*; *Gerber et al., 2021*).

Despite the established benefits of MVPA, the 2018 and 2022 India Report Cards, national assessments of physical activity, found a large proportion of children and youth do not meet the recommended MVPA levels in India (50% of children and youth reported less than 60 min of daily MVPA in 2021) (*Bhawra et al., 2018*; *Bhawra et al., 2023a*). Promoting active school transportation (AST), including biking or walking to school can potentially increase MVPA levels and reduce the prevalence of non-communicable diseases as AST addresses various interconnected aspects of health and well-being, ranging from physical health to mental well-being (*Stark et al., 2018*; *Drake et al., 2012*; *Sun, Liu & Tao, 2015*). For instance, school-aged children in China who engaged in AST were found to be

associated with lower odds of obesity and depressive symptoms (*Sun, Liu & Tao, 2015*). Another study examining AST and physical activity in children from across 12 countries, including India, found that in most countries, 9 to 12-year-old children engaging in AST were more likely to meet the recommended MVPA guidelines of at least 60 min per day compared to their counterparts who did not engage in AST (*Denstel et al., 2015*). Moreover, AST has also been associated with a range of cognitive benefits, including 30% higher odds of perceiving high academic performance and reading competency (*Jussila et al., 2023*).

Despite studies indicating that a large proportion of children and youth in India currently walk or bicycle to school at least once per week (52% in Chennai, India and 63% in Hyderabad, India) (*Kingsly et al., 2020*; *Tetali, Edwards & Roberts, 2016*), growing environmental barriers may negatively impact child and youth engagement in AST. For instance, a previous study found that children and youth who perceived an air pollution problem were 38.3% less likely to engage in AST (*Bhawra et al., 2023b*). Moreover, this era of rising climate emergencies is making it increasingly challenging to engage in outdoor physical activities (*Bhawra et al., 2023b*; *Obradovich & Fowler, 2017*; *Patel et al., 2023*). As a result, there is a need to understand AST in India—one of the fastest-growing countries, where sustainable development is needed to link active transportation with climate change mitigation (*Sciubba, 2023*).

Currently, there is limited evidence in India, particularly from multiple centres with varied population sizes—an important gap given India's wide variation in population centres (*Bhawra et al., 2023a*). This evidence gap is even more pronounced when it comes to child and youth perceptions of the environment. Although one study directly examined AST and MVPA among children in Bangalore, India (*Denstel et al., 2015*), no studies have incorporated the role of community and environment perceptions, as well as compared multiple models across various sociodemographic variables including rural and urban regions. *Kingsly et al. (2020)* previously found that perceptions of various environmental factors including dangerous crossings, crime rates, and traffic, were associated with lower odds of engaging in AST among children and youth in Chennai, India; however, their study was limited to one population centre and did not examine how environmental factors played a role in the relationship between AST and MVPA (*Kingsly et al., 2020*). To address this critical evidence gap, this study aims to understand the association between AST and MVPA among children and youth in several population centres in India while incorporating their own perspectives of the environment and capturing variations across geographic regions (rural *vs* urban), age, and gender.

## METHODS

### Design

As a part of a multi-centre study, data for this cross-sectional, observational study were collected from children and youth aged 5 to 17 through online surveys during the coronavirus disease lockdown in India in 2021. Based on the sample size calculation for proportions, the study needed 385 participants to achieve a 95% confidence level and 5% margin of error—our study exceeded the minimum sample size required by surveying

1,042 children and youth. Prior to data collection, ethics approval was obtained from the Ethics Committee of Jehangir Clinical Development Centre Pvt. Ltd in Pune, Maharashtra (EC registration number—ECR/352/Inst/MH/2013/RR-19).

### Recruitment and participants

Recruitment of participants for this study was conducted using a multi-stage stratified random sampling method. After randomly selecting five Indian states (Maharashtra, Gujarat, Telangana, Madhya Pradesh, and Tamil Nadu), 28 cities (*i.e.,* 14 urban regions) and nearby villages (*i.e.,* 14 rural regions) were randomly selected. Thereafter, a list of 50 schools was generated from these regions and the school principals for each of the schools were contacted with study information. The number of schools involved in each region varied based on the responses from school principals, with 41 principals agreeing to participate. These schools then electronically distributed study information and consent forms to students and parents. Online informed consent was obtained from all students and their parents before digital survey links were emailed to children and youth. All surveys were completed individually and anonymously, with children and youth under 13 years receiving assistance from their parents. The survey captured sociodemographic characteristics, active living behaviours, and perceptions of their environment.

## MEASURES

### Sociodemographic characteristics

The survey collected data from participants on age, gender (boy *vs* girl), and geographic location (rural *vs* urban). The study determined the age of participants by asking them to report their date of birth, which was categorized into the aged 5 to 12 group or the aged 13 to 17 group. Gender was ascertained by asking children and youth to specify whether they identified as 'boy' or 'girl', while their geographic location was determined by assessing the distance of the schools from various population centres across five Indian states (*Vispute et al., 2023*; *Khadilkar et al., 2022*).

### Physical activity

Children and youth were asked to report the weekly frequency and duration they engaged in a list of activities as a part of an adapted version of the IPAQ short-form questionnaire (S1 Appendix) (*Craig et al., 2003*). Each activity was categorized by intensity as inactivity, light activity, or MVPA. The average duration spent on MVPA daily was calculated by combining the time spent on activities that were categorized as MVPA. Total weekly MVPA was calculated by multiplying the duration in minutes spent per day and the frequency of each activity per week for each individual. Thereafter, total weekly MVPA was divided by seven to generate the average daily minutes of MVPA. The intraclass coefficient for self-reported MVPA among children and youth was found to be fair-to-good by previous research (*Ng et al., 2019*; *Strugnell et al., 2014*).

### Active school transportation

Participation in AST was determined by asking children and youth "Do you bike to school and "Do you walk to school?", with dichotomous responses (yes or no). A new variable

was created in which participants who biked to school, walked to school, or both biked and walked to school were considered to engage in at least one form of AST. Self-reported AST has been previously studied by *Larouche et al. (2014a)* who conducted a systematic review and found that self-reports for AST by children and youth (aged 5 to 17) had substantial to perfect test-retest reliability and convergent validity.

## Perceptions of the environment

This study examined perceptions of the environment by providing statements and asking participants to select the answer that best applied to them and their neighbourhood. Participants were provided with a 5-point Likert scale (strongly disagree, somewhat disagree, neither agree nor disagree, somewhat agree, strongly agree). The responses were dichotomized to those who agreed (somewhat agree and strongly agree) and those who disagreed (somewhat disagree, strongly disagree) with statements including "There is a high crime rate in my neighbourhood" and "Air pollution prevents me from being active outside."

## Peer support

Peer support was determined by asking participants about the number of close friends they had who were physically active with response options ranging from "zero" to "four or more". Response options that were "zero" were converted to "has no active friends", while response options that were greater than zero were converted to "has one or more active friends".

## Statistical analysis

This study utilized R studio to conduct all analyses (*R Studio Team, 2015*). The primary independent variable for this study was AST, while the dependent variable was the average daily minutes spent on MVPA. The sample was stratified by gender (boys and girls), location (rural and urban), and age group (aged 5 to 12 and aged 13 to 17). Binary variables such as gender and location were coded as 0 and 1 to represent the respective groups (*e.g.*, girls = 0, boys = 1), allowing the models to estimate differences between these categories (*Schneider, Hommel & Blettner, 2010*; *Lunt, 2015*). Before proceeding with regression analyses, the normality of residuals for the MVPA data was assessed using visual inspection of Q-Q plots. Despite slight deviations from normality, we proceeded with parametric linear regression models, as previous research indicates that these models are robust to non-normality when sample sizes are sufficiently large (N >30 or 40) (*Ghasemi & Zahediasl, 2012*; *Schmidt & Finan, 2018*; *Ernst & Albers, 2017*). Chi-square tests were conducted to compare the engagement in AST across sociodemographic groups. Thereafter, seven linear regression models were developed: the overall and age models were adjusted for age, gender, and location; the gender models were adjusted for age and location; and the location models were adjusted for age and gender. The results of this study were deemed statistically significant at a threshold $p$-value less than 0.05.

**Table 1 Children and youth sample summary by gender, age, and location.**

|  | Total ($N = 992$) | |
| --- | --- | --- |
| **Sample Characteristics** | **N** | **%** |
| **Age Group** | | |
| 5 to 12 years | 547 | 55.1 |
| 13 to 17 years | 445 | 44.9 |
| **Gender** | | |
| Boys | 499 | 50.3 |
| Girls | 493 | 49.7 |
| **Location** | | |
| Rural | 400 | 40.3 |
| Urban | 592 | 59.7 |

## RESULTS

Of the 1,042 Indian children and youth who participated in this study, 992 children and youth aged 5 to 17 provided data on relevant indicators and were included in the analyses. The findings in Table 1 show the summary statistics of the participants in the sample. A larger proportion (55.1%) of children and youth were aged 5 to 12, while 44.9% were aged 13 to 17. The gender distribution was relatively balanced, with 50.3% of the participants being boys and 49.7% being girls. More children and youth (59.7%) resided in urban than rural regions (40.3%).

Table 2 shows the average daily minutes spent on MVPA and the proportion of children and youth who engaged in AST. Children and youth spent 82.5 min on MVPA, with the aged 5 to 12 group spending 87.9 min and the aged 13 to 17 group spending 95.9 min. When examining gender, boys spent 107.6 min on MVPA, while girls engaged in 57.8 min. Furthermore, urban residents engaged in 83.4 min of MVPA per day, while rural residents engaged in 81.2 min. A majority of children and youth overall did not engage in AST (66.0%). In particular, there was a significant difference in the proportion of boys (37.3%) that engaged in AST compared to 30.6% of girls ($p = 0.030$). In terms of geographic location, there was a significant difference in the proportion of urban residents (14.7%) who engaged in AST and rural residents (62.5%) who engaged in AST ($p < 0.001$). There were also variations across age groups with 28.9% of the aged 5 to 12 group engaging in AST and 40.8% of the aged 13 to 17 group engaging in AST ($p < 0.001$).

The multiple linear regression model for the overall sample demonstrating the association between engaging in AST and MVPA is shown in Table 3. Children and youth who engaged in AST were associated with higher MVPA than those who did not engage in AST ($\beta = 18.020$, 95% CI [5.890 to 30.149], $p = 0.004$). Further, in terms of neighbourhood safety perceptions, reporting high crime in the neighbourhood was associated with lower MVPA ($\beta = -21.921$, CI [$-36.195$ to $-7.647$], $p = 0.003$) than those who did not report high crime. Reporting that air pollution prevents them from being active outside was associated with lower MVPA than those who did not perceive that air pollution prevents them ($\beta = -12.472$, CI [$-23.797$ to $-1.147$], $p = 0.031$). When

**Table 2** Average moderate to vigorous physical activity (MVPA) levels by engagement in active school transportation, across age, gender, and location.

| Category | MVPA<br><br>Average minutes per day (SD) | Active school transportation: Yes<br>n (%) | Active school transportation: No<br>n (%) | Chi-squared test statistic (*p*-value) |
|---|---|---|---|---|
| Aged 5 to 12 | 87.9 (89.8) | 153 (28.9) | 376 (71.1) | 14.5 (*p* < 0.001) |
| Aged 13 to 17 | 95.9 (103.0) | 181 (40.8) | 263 (59.2) | |
| Boys | 107.6 (80.7) | 186 (37.3) | 312 (62.7) | 4.7 (*p* = 0.030) |
| Girls | 57.8 (65.0) | 151 (30.6) | 342 (69.4) | |
| Urban | 83.4 (79.4) | 87 (14.7) | 504 (85.3) | 240.5 (*p* < 0.001) |
| Rural | 81.2 (74.0) | 250 (62.5) | 150 (37.5) | |
| Overall | 82.5 (77.3) | 337 (34.0) | 654 (66.0) | |

**Table 3** Association of active school transportation, perception of crime, air pollution, and peer support with moderate to vigorous physical activity (MVPA).

| Outcome variable—Minutes of Moderate to Vigorous Physical Activity | Beta coefficients (95% Confidence Intervals)[a] |
|---|---|
| Active school transportation: Yes | 18.020[*] (5.890, 30.149) |
| No –(Ref) | |
| Disagree –(Ref) | |
| There is a high crime rate in my neighborhood: Agree | −21.921[*] (−36.195, −7.647) |
| Disagree –(Ref) | |
| Air pollution prevents me from being active outside: Agree | −12.472[*] (−23.797, −1.147) |
| Disagree –(Ref) | |
| Has one or more active friends: Yes | 32.391[*] (9.264, 55.518) |
| No –(Ref) | |
| Constant | 18.577 (−7.375, 44.530) |

Notes.

[*] Indicates a statistically significant relationship at the *p* < 0.05 level.

[a] The model was also adjusted for age, gender, and location.

examining peer support, having one more active friends was associated with higher MVPA levels ($\beta = 32.391$, CI [9.264 to 55.518], $p = 0.006$) than those who had no active friends.

The regression model examining the association between engaging in AST and MVPA levels across various age groups is shown in Table 4. Children and youth aged 13 to 17 who engaged in AST were associated with higher MVPA levels ($\beta = 23.645$, CI [6.265 to 41.026], $p = 0.008$) than those who did engage in AST; however, this relationship was not significant in the aged 5 to 12 group. In the aged 13 to 17 group, agreeing to a high crime rate in the neighbourhood was associated with lower MVPA levels ($\beta = -24.365$, CI [−46.053 to −2.677], $p = 0.028$) than those who disagreed. Further, having one or more active friends

**Table 4** Association of active school transportation, perception of crime, air pollution, and peer support with Moderate to vigorous physical activity (MVPA) in various subgroups.

| | Outcome variable – Minutes of Moderate to Vigorous Physical Activity | | | | | |
|---|---|---|---|---|---|---|
| | Beta Coefficients (95% Confidence Intervals) | | | | | |
| | Age group (years) | | Gender | | Geographic location | |
| | 5 to 12[a] | 13 to 17[a] | Boys[b] | Girls[b] | Urban[c] | Rural[c] |
| Active school transportation: Yes | 3.639 (−14.310, 21.588) | 23.645* (6.265, 41.026) | 31.746* (13.847, 49.645) | 4.081 (−11.677, 19.839) | 15.894 (−6.873, 38.661) | 19.480* (5.364, 33.596) |
| No –(Ref) | | | | | | |
| There is a high crime rate in my neighborhood: Agree | −16.733 (−35.533, 2.067) | −24.365* (−46.053, −2.677) | −15.793 (−39.197, 7.611) | −26.503* (−43.063, −9.942) | −17.386 (−38.611, 3.839) | −26.576* (−46.223, −6.930) |
| Disagree –(Ref) | | | | | | |
| Air pollution prevents me from being active outside: Agree | −19.642* (−34.403, −4.880) | −5.644 (−23.102, 11.814) | −17.757 (−35.558, 0.043) | −3.780 (−20.643, 13.083) | −13.002 (−29.015, 3.010) | −11.118 (−27.497, 5.260) |
| Disagree –(Ref) | | | | | | |
| Has one or more active friends: Yes | 27.035* (0.674, 53.396) | 41.641* (−1.766, 85.047) | 6.582 (−42.577, 55.741) | 43.577* (20.240, 66.913) | 37.118* (11.061, 63.175) | 2.259 (−71.731, 76.248) |
| No –(Ref) | | | | | | |
| Constant | −12.773 (−58.269, 32.722) | 49.892 (−62.675, 162.459) | 28.496 (−31.244, 88.236) | 16.159 (−22.289, 54.607) | 4.848 (−34.964, 44.660) | 16.472 (−61.965, 94.909) |

**Notes.**
*Indicates a statistically significant relationship at the $p < 0.05$ level.
[a]The model was also adjusted for age, gender, and location.
[b]The models were adjusted for age and location.
[c]The models were adjusted for age and gender.

was associated with higher MVPA levels in the aged 5 to 12 group ($\beta = 27.035$, CI [0.674 to 53.396], $p = 0.006$). Similarly, reporting that air pollution prevents them from being active outside was associated with lower MVPA in the aged 5 to 12 group but not the aged 13 to 17 group ($\beta = -19.642$, CI [−34.403 to −4.880], $p = 0.009$)

The association between AST and MVPA levels in the gender models is also shown in Table 4. For boys, AST was associated with higher MVPA levels than those who did not engage in AST ($\beta = 31.746$, CI [13.847 to 49.645], $p < 0.001$); however, this association was not significant for girls. For girls, agreeing to a high crime rate in the neighbourhood was associated with lower MVPA levels than disagreeing ($\beta = -26.503$, CI [−43.063 to −9.942], $p = 0.002$). In contrast, for girls, having one or more active friends was associated with higher MVPA levels than those who had no active friends ($\beta = 43.577$, CI [20.240 to 66.913], $p < 0.001$), but this association was not significant for boys.

In terms of geographic location, among children and youth living in rural centres, AST was associated with higher MVPA levels ($\beta = 19.480$, CI [5.364 to 33.596], $p = 0.007$); however, this association was not significant in urban centres. Similarly, the perception of high crime in the neighbourhood was associated with less MVPA in the rural model ($\beta$

$= -26.576$, CI [$-46.223$ to $-6.930$]), $p = 0.008$), but not the urban model. In contrast, having one more active friends was significantly associated with higher MVPA levels than having no active friends in the urban model ($\beta = 37.118$, CI [11.061 to 63.175], $p < 0.006$), but not in the rural model.

## DISCUSSION

This study's main finding was that engaging in AST was associated with higher MVPA among children and youth. This highlights its potential to contribute to reducing the NCD burden, owing to MVPA's established role as a key preventative factor for NCDs among school-aged children and youth (*Poitras et al., 2016*). When incorporated into children and youth's daily lives, physical activity interventions such as AST may make it easier for them to be active (*Canada PHA, 2018*; *Jones et al., 2019*), potentially providing a sustainable strategy for promoting physical activity. Our study findings align with the growing body of evidence in other countries indicating that children and youth in low-and middle-income countries who engaged in AST were more active and accumulated more physical activity (*Jones et al., 2019*; *Larouche et al., 2014b*; *Reiner et al., 2013*; *Schoeppe et al., 2013*; *Prince et al., 2022*); however, unlike our study, these studies have not specifically examined MVPA. Moreover, although *Denstel et al. (2015)* found that AST was linked to a higher likelihood of meeting the recommended MVPA guidelines of 60 min per day among children and youth in Bangalore, India, our results add to the existing evidence by capturing AST and MVPA across various locations (28 cities and villages), including rural and urban regions.

The importance of promoting AST becomes even more significant in the context of the global physical inactivity pandemic, which is a growing concern as it has only been linked to several NCDs among youth (*Gore et al., 2011*). This issue disproportionately impacts individuals living in low-and middle-income countries, who already experience a greater burden of NCDs (*Ndubuisi, 2021*; *Katzmarzyk et al., 2022*; *Swinburn et al., 2019*). Moreover, the increasing frequency of climate emergencies including floods, droughts, and poor air quality warnings (*Ali, Modi & Mishra, 2019*), further complicate efforts to increase physical activity by making outdoor physical activity more difficult (*Bhawra et al., 2023b*; *Obradovich & Fowler, 2017*; *Patel et al., 2023*). Given these challenges, holistic solutions such as promoting AST are critical for addressing the combined effects of physical inactivity, climate emergencies, and widening inequities in NCDS in countries like India (*WHO, 2023*; *Ding & Elbarbary, 2021*; *Romanello et al., 2023*).

Our study also provides unique insights by examining how the relationship between AST and MVPA varies based on child and youth perceptions of the environment. Previous studies conducted among youth have found that crime incidence in youths' neighbourhoods was associated with a lower likelihood of engaging in physical activity (*Robinson, Carnes & Oreskovic, 2016*; *Janssen, 2014*). However, it is imperative to consider child and youth perceptions because in some cases, the perception of the environment may have a differential impact on active living behaviours compared to objective environmental factors. For instance, a previous study found that the perception of
crime in the neighbourhood among 11-to-15-year old's was a stronger predictor of physical activity than the actual crime rates (*Janssen, 2014*). Similarly, a meta-analysis conducted by Rees-Punia et al. found that participants feeling safe from crime had 27% higher odds of engaging in physical activity (*Rees-Punia, Hathaway & Gay, 2018*). In line with these findings, our study found that the perception of high crime in the neighbourhood was associated with lower MVPA in the overall sample (*Poitras et al., 2016*; *Janssen & LeBlanc, 2010*; *Gerber et al., 2021*).

This study also found that perceptions of air pollution were associated with lower MVPA. This perception was specific to children and youth limiting their outside time due to air pollution, which adds to existing evidence that Indian younger children and youth who perceive air pollution to be a problem accumulate lower MVPA (*Patel et al., 2023*). Less time spent outside is directly related to lower AST, which ultimately results in lower MVPA. These findings could be attributable to the independent mobility of children and youth and the perceptions of their parents which might limit their ability to minimize exposure to negative environmental factors, including air pollution (*Schoeppe et al., 2013*; *Pickhardt, 2010*).

As child and youth AST as well as MVPA vary across age cohorts, gender, and more importantly type of environment (rural *vs.* urban) (*Bhawra et al., 2023b*; *Patel et al., 2023*; *Carver et al., 2011*; *Marzi et al., 2020*), this study also provides unique insights by introducing seven models that compare the associations between AST and MVPA across age, location, and gender. This new evidence is important to understand active living patterns in India because thus far, no previous research on children and youth in India has investigated the relationship between AST and MVPA levels across such cohorts (*Bhawra et al., 2023a*; *Denstel et al., 2015*). Our study contributes to bridging this gap by depicting that while AST was associated with MVPA among children and youth aged 13 to 17, it was not associated with MVPA among children and youth aged 5 to 12. Additionally, a lower proportion of children and youth aged 5 to 12 engaged in AST compared to children and youth aged 13 to 17. These findings corroborate existing evidence suggesting that older adolescents have more independent mobility, which may play a role in their active transportation patterns and physical activity levels (*Schoeppe et al., 2013*; *Pickhardt, 2010*).

On the contrary, older adolescents are also more likely to not only witness but also experience serious forms of violence, including from crime in the community (*Finkelhor et al., 2015*), which may also influence them to modify their behaviours more than younger children and youth in response to high crime in their neighbourhood. For instance, this study found that high crime was associated with lower MVPA in the aged 13 to 17 group, but not in the aged 5 to 12 group. The differential impact of perceived crime rates on MVPA between these age groups suggests that neighbourhood safety concerns stemming from perceived crime rates might influence older children's engagement in physical activity. In contrast, younger children often rely more heavily on adult supervision (*Buliung et al., 2017*; *Ghekiere et al., 2017*; *Riazi et al., 2019*), potentially making them less likely to be influenced by concerns about perceived crime rates.

In addition to age-related differences, this study also highlighted variations in the relationship between AST and MVPA in children and youth living in rural *vs* urban areas.

We found that AST was associated with higher MVPA among children living in rural regions of India, but not urban regions, a distinction that has not been captured by previous studies. These findings may corroborate existing evidence that rural Indian children accumulate more AST and MVPA (*Bhawra et al., 2018*; *Bhawra et al., 2023a*; *Katapally et al., 2016*). As studies have found traffic to be a barrier to active transportation (*Wilson, Clark & Gilliland, 2018*), it is plausible that the positive association between active transportation and MVPA in rural regions could be attributed to fewer barriers to physical activity, including less traffic congestion as seen in higher density and populated urban cities (*Akbar et al., 2023*). As this study did not capture objective environmental barriers such as traffic levels, further research is required to better understand the barriers to active transportation in urban areas. Moreover, rural regions may allow more opportunities for unstructured outdoor play and exploration, which is important because previous evidence indicates that these factors can have an important role in promoting physical activity (*Dankiw et al., 2020*). Our study findings highlight the need for tailored interventions that address the distinct challenges and opportunities to MVPA presented by rural and urban environments (*Chandler et al., 2019*).

Gender-based disparities in AST were also identified in this study. In particular, AST was associated with MVPA in boys, but not in girls, which may be attributed to cultural expectations of girl engagement in AST and physical activity in India (*Mathews et al., 2016*). Safety concerns may disproportionately impact girls engaging in AST compared to boys, contributing to its minimal impact on MVPA levels. For instance, a study conducted in Chennai, India found that many girls experienced safety concerns and fear of gender-based violence when engaging in active transportation in their neighbourhood (*Adlakha, Hipp & Brownson, 2016*). Due to higher safety concerns, AST may not promote MVPA among girls, despite it promoting MVPA among boys. Moreover, this study also reinforces the impact of girls' safety concerns on their activity levels as this study found that among girls exclusively, a high crime rate was associated with lower MVPA levels. These gender-specific findings highlight that recognizing and addressing gender-based disparities is essential for fostering an inclusive environment that encourages both boys and girls to embrace active modes of transportation and promote physical activity.

The study also reinforces the findings of previous literature which emphasize the importance of active friends for promoting physical activity (*Jago et al., 2011*; *Leggett et al., 2012*), as having one or more active friends was associated with higher MVPA levels. Similarly, when examining age-specific dynamics, having active friends was associated with higher MVPA only in the aged 5 to 12 group, which highlights the pivotal role of social interactions and friendships on activity behaviours, particularly among younger children and youth. In terms of geographic location, having active friends was associated with MVPA among children and youth living in urban centres exclusively, highlighting that the impact of peer support on active living behaviours may differ based on geographic location; however, further research is needed to unpack the mechanisms and factors contributing to these disparities in India. Based on our study findings, interventions that promote peer-relationship building and active social circles could have a pronounced impact on the

physical activity levels of children and youth, especially if they take into consideration the varying dynamics of different sociodemographic groups of children and youth.

## Strengths and limitations

This study not only captured child and youth physical activity levels but also examined their engagement in AST and perceptions of the environment across 28 different cities and villages in India, which has not been achieved by previous studies. However, future studies should aim to include a more representative sample to enhance the generalizability of the findings. Furthermore, given the role of weather conditions on active living behaviours, future longitudinal studies should link objective weather data with AST and MVPA to account for variations attributable to seasonality (*Katapally, Rainham & Muhajarine, 2015*; *Katapally, Rainham & Muhajarine, 2016*; *Katapally & Muhajarine, 2015*). Additionally, as active transportation was a binary measure, we were not able to understand the nuanced variations in AST. Capturing AST engagement using ecological momentary assessments could be used in future research to not only account for differences in AST throughout the week and across different places but also to reduce recall biases (*Ibrahim, Hammami & Katapally, 2023*; *Katapally & Chu, 2020*). Another limitation of this study is that MVPA was measured using self-reports from children and youth which may differ from objectively measured MVPA. Additionally, surveys were completed in various settings, with children under 13 receiving assistance from their parents, which may have introduced variability in responses and potential parental influence. Future studies should examine the relationship between AST and objectively measured MVPA among children and youth through ubiquitous devices such as smartphones (*Poitras et al., 2016*; *Katapally, Rainham & Muhajarine, 2016*; *Deng & Fredriksen, 2018*). However, as ethically leveraging youth-owned data is critical, digital citizen science approaches can be used to maximize the potential of youth ownership of smartphones in India (*Katapally, 2019*; *Katapally, Hammami & Chu, 2021*).

## CONCLUSION

This is the first study conducted across multiple centres, including both rural and urban locations, that examined the association between AST and MVPA among children and youth in India. The study findings indicate that engaging in AST can promote MVPA among children and youth in India. This study also provides unique insights by highlighting that the relationship between AST and MVPA varies across age, gender, and rural *vs* urban residence. Due to the population size and geographic and cultural diversity of India, it is imperative to substantiate this evidence with future studies that include a representative sample of Indian children and youth.

### Funding

This study is supported by the Canada Research Chairs Program, which funds Dr. Tarun Katapally's research program. The funders had no role in study design, data collection and analysis, decision to publish, or preparation of the manuscript.

### Grant Disclosures

The following grant information was disclosed by the authors:
The Canada Research Chairs Program.

### Competing Interests

The authors declare there are no competing interests.

### Author Contributions

- Tarun Reddy Katapally conceived and designed the experiments, performed the experiments, authored or reviewed drafts of the article, and approved the final draft.
- Jamin Patel analyzed the data, prepared figures and/or tables, authored or reviewed drafts of the article, and approved the final draft.
- Anuradha Khadilkar conceived and designed the experiments, performed the experiments, authored or reviewed drafts of the article, and approved the final draft.
- Jasmin Bhawra conceived and designed the experiments, performed the experiments, authored or reviewed drafts of the article, and approved the final draft.

### Human Ethics

The following information was supplied relating to ethical approvals (*i.e.*, approving body and any reference numbers):

Ethics Committee of Jehangir Clinical Development Centre Pvt. Ltd in Pune, Maharashtra

### Data Availability

The raw data is available at Figshare: Patel, Jamin (2024). AT MVPA Raw Data. figshare. Dataset. https://doi.org/10.6084/m9.figshare.25664589.v2.

### Supplemental Information

Supplemental information for this article can be found online at http://dx.doi.org/10.7717/peerj.18350#supplemental-information.

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
