# Peer review of "The critical need for child and youth perceptions of active living in India: capturing context complexity in rural and urban regions"

_PeerJ, doi:10.7717/peerj.18350_

## Round 0.1 · original submission · Minor Revisions

Thank you for your submission. The reviewers have identified a number of minor concerns that should be addressed.

Reviewer 1 ·

Basic reporting

1. Please mention how the MVPA data were collected in the Abstract (e.g., using IPAQ) to avoid confusion with objective measurements. Alternatively, include a blanket statement at the beginning that all data for this study were collected using surveys.
2. Consider if "association between AST and MVPA" is the appropriate terminology, given other aspects (perception of crime and air pollution, and peer support) are reported. This also applies to Tables 3 and 4: due to the title "engaging in ATS and MVPA" appears to be incomplete given the tables include crime, air pollution perception, and active friends. Consider revising the title to reflect a technically accurate description of the findings.
3. Chi-square tests are not mentioned in the statistical analysis description, despite being reported. Please complete the reporting of this statistical analysis.
4. The organization of information in Table 2 and the description in Lines 221-233 should follow a similar order: age category first, then gender, and geographic location. This will ensure consistency with the Abstract and Table 4.
5. (Line 77) The World Health Organization recommendation should be stated as "at least an average of 60 minutes MVPA" to ensure technical accuracy and following the methods of analysis in this study.
6. I suggest to begin the Discussion with a summary of the main findings, rather than repeating the rationale of the study.

Experimental design

(Line 147-150) It appears that the survey is completed by the children and administered by their parents at home. How is data validation ensured in this process? Consider mention this as a limitation, specifying how the survey ensures it is completed by the intended individual without supervision from the research team.

Validity of the findings

How is MVPA, derived from the questionnaire, justified in relation to AST, given it includes various school and after-school activities (Line 167-172)? For instance, how does walking or cycling to school relate to activities like weight training or swimming (examples of the moderate and vigorous activity in the questionnaire)? Please explain why you used this data.

Additional comments

I commend the authors for conducting this highly interesting study and contributes to the literature on physical activity in youth. Overall writing of the manuscript is clear and professional. However, framing it solely as an examination of the association between ATS and MVPA is incomplete and potentially misleading. The study also explores perceptions of crime, air pollution, and peer support for MVPA, which should be highlighted in the title and abstract. While ATS is the primary independent variable, as mentioned in the Methods, MVPA as the dependent variable would be more appropriate as the main focus of the reporting, especially in the titles of the tables.

Reviewer 2 ·

Basic reporting

1. The language used is clear. However, I would like to suggest that the author use the term boys and girls instead of male and female as the reference are children. Also, for line 158, suggest to use the term categorized instead of sorted.
2. Literature references are sufficient. However, I would like the authors to relook at the reference referred to as [8] in line 78. I think it was references wrongly.

Experimental design

On the methodology,

1. Sample size was calculated to be 385 participants, however, the study has a total sample of 1042 children. May I know why the final sample size is so much larger than needed?

2. For recruitment, it is stated that 28 cities, urban and rural were selected. Can the author please state the exact number of rural and urban location selected?

3. Data were collected via an online questionnaire that was emailed to the children. The ages of the participants ranges from 5 years to 17 years. Can the author please explain how they ensure that the responses from younger children are valid as they may not be able to read effectively to answer the questions.

4. The title eluded to measuring built environment, but built environment was not measured and was never mentioned in the article. For the perceptions of the environment, please state all aspect that was measured under this. Was it only perceived crime rate and air quality?

Validity of the findings

Results:

1. Line 235 stated that multiple linear regression model was used to test association between engaging in AST and MVPA (Table 3). However, Multiple linear regression requires all variables to be continuous data. Please explain the use of multiple linear regression in this case. Or did the authors used Logistic regression? Please clarify.

Additional comments

Overall, the paper was clearly written with some clarification needed in certain areas.

Annotated reviews are not available for download in order to protect the identity of reviewers who chose to remain anonymous.

---

## Round 0.2 · Minor Revisions

Thank you for your revised manuscript. One of the prior reviewers has a suggestion and a question for you to address.

Reviewer 2 ·

Basic reporting

The article was clearly written with professional English used throughout. Literature references were adequate.

Experimental design

The article is an original research within the aims and scope of the journal. Research question was well defined, relevant & meaningful. It is stated how research fills an identified knowledge gap. Methods described with sufficient detail & information to replicate.

Validity of the findings

Impact and novelty not assessed. Meaningful replication encouraged where rationale & benefit to literature is clearly stated. Conclusions are well stated, linked to original research question & limited to supporting results

Additional comments

Dear authors,

Thank you for your clear responses to my query. However, I do have one more suggestion. Please consider including a reference for your use of linear regression with binary variable.

I also have one new query. Can you clarify if you checked whether the PA data were normally distributed. In general, PA data are not normally distributed. Please clarify.

---

## Round 0.3 · accepted · Accept

Thank you for your revised submission. I am satisfied that you have addressed the remaining concerns of the reviewer, and am happy to accept your paper for publication.